# *ENTP:* Encoder-only Next Token Prediction

**Ethan Ewer**[*]                        *eewer@wisc.edu*
*University of Wisconsin-Madison*

**Daewon Chae**[*]                       *cdw098@korea.ac.kr*
*Korea University*

**Thomas Zeng**[*]                       *tpzeng@wisc.edu*
*University of Wisconsin-Madison*

**Jinkyu Kim**                       *jinkyukim@korea.ac.kr*
*Korea University*

**Kangwook Lee**                     *kangwook.lee@wisc.edu*
*University of Wisconsin-Madison*

**Reviewed on OpenReview:** *https://openreview.net/forum?id=CGHi289y8e*

## Abstract

Next-token prediction is conventionally done using decoder-only Transformers with causal attention, as this approach allows for efficient reuse of keys and values. What if we were not compute-limited, should we still use decoder-only Transformers? In this work, we introduce **E**ncoder-only **N**ext **T**oken **P**rediction (ENTP). We explore the differences between ENTP and decoder-only Transformers in expressive power and complexity, highlighting potential advantages of ENTP in settings with unbounded compute. We introduce the Count3 task and show, both theoretically and experimentally, that while ENTP can perform this task easily, a decoder-only Transformer cannot. Finally, we empirically demonstrate the superior performance of ENTP across representative tasks where next-token prediction based Transformers can be evaluated, including addition, in-context learning, and language modeling.

## 1 Introduction

Traditionally, auto-regressive language modeling has relied on decoder-only Transformers (Vaswani et al., 2017) with causal attention, trained using the next-token prediction objective. Causal attention ensures that each token can only attend to previous tokens, preventing future tokens from influencing past outputs. This mechanism makes training and inference more efficient, as past keys and values do not need to be recomputed for each token. This efficiency enables the scaling of decoder-only Transformers, such as GPT-4 (Achiam et al., 2023) and Llama-3 (Dubey et al., 2024), up to billions of parameters using current hardware.

However, causal attention also introduces artificial constraints. Given tokens $x_1, x_2, ..., x_n$, the contextual embedding of $x_j$ (where $j < n$) can only attend to embeddings of earlier tokens, even when predicting $x_{n+1}$. While this constraint ensures a strict causal structure, it may not always be necessary or beneficial. We investigate what happens when we remove this constraint, while still maintaining causality externally.

Specifically, we look at Encoder-only Transformers, which are typically used for tasks like classification, and do not impose this causality constraint. Though traditionally not used for auto-regressive tasks, encoder-only architectures can be adapted for next-token prediction. When computing the output at the current time step, an encoder-only Transformer, or any sequence model, can be made causal by only providing inputs up

---

[*]Equal contribution.

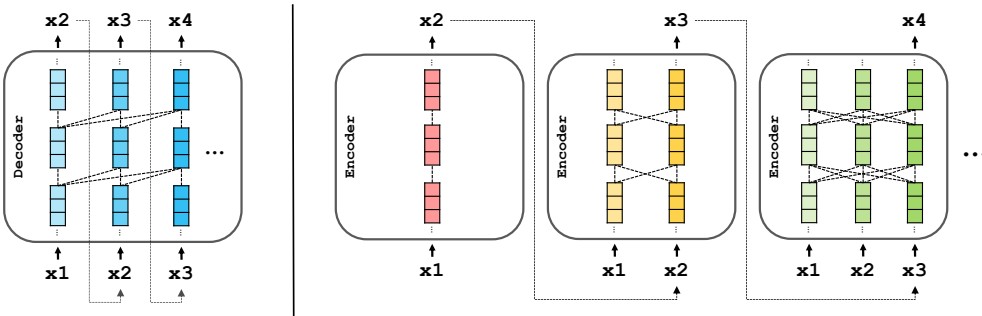

Figure 1: **Decoder-only vs. Encoder-only Transformers in next token prediction.** Decoders use causal attention, ensuring that each token attends only to the preceding tokens. In contrast, encoders allow all tokens to attend to each other by performing attention computation from scratch for each token prediction.

to and including the current time step (see Figure 1). Therefore, in this work, we investigate the idea of using encoder-only Transformers for next-token prediction. We summarize our findings below.

**Functions expressible with Decoder-only and Encoder-only Transformers.** We demonstrate that the sets of functions expressible by decoder-only and encoder-only Transformers are not comparable, which goes against intuition that the expressivity of encoders would subsume that of decoders. Rather, there exist functions expressible with decoder-only Transformers that are not expressible with encoder-only Transformers, and vice versa, as well as functions expressible by both architectures.

**Complexity of Decoder-only and Encoder-only Transformers.** Based on the minimum time and space complexities, we give a description of the functions that can be performed by decoder-only and encoder-only Transformers. We propose an auto-regressive task that can be performed by encoder-only Transformers, and cannot be performed by decoder-only Transformers (given that some mild assumptions hold). We validate our hypothesis with small experiments using small decoder-only and encoder-only Transformers, as well as experiments fine-tuning GPT-4o (Achiam et al., 2023), Llama3-8B (Dubey et al., 2024), and BERT (Devlin et al., 2019).

**Experiments on Small-Scale Language Modeling Tasks.** We compare the performance of decoder-only and encoder-only Transformers on a range of small-scale language modeling tasks. We test the sample complexity and length generalization capabilities of decoders and encoders using arithmetic tasks (Lee et al., 2023). We also train both models to perform in-context learning (Garg et al., 2022) on various simple functions. Additionally, we train small decoder-only and encoder-only Transformers on a text dataset (Gokaslan et al., 2019) and assess their performance on language modeling tasks.

## 2 Related Work

**Expressive Power of Transformers.** There have been various literature exploring the expressive power of Transformers. From the lens of universal approximation, Yun et al. (2020) showed that any continuous sequence-to-sequence function over a compact set can be approximated arbitrary close by a Transformer (of finite albeit very large size). Other works approach expressiveness from the perspective of computability and complexity such as Pérez et al. (2021) which showed Transformers are Turing complete and Merrill et al. (2022); Merrill & Sabharwal (2024); Li et al. (2024) which use circuit complexity to characterize the languages recognizable by Transformers of fixed depth. Giannou et al. (2023) presents a framework for Transformers as universal computers by placing them in a loop. Communication complexity has also been used to show the impossibility of one-layer Transformers of expressing certain functions e.g. induction head, without the

model size being linear in the length of the input (Sanford et al., 2024b;a). We note that the existing bounds (Sanford et al., 2024b;a) are highly related, but they do not directly imply the relative expressive power of encoder and decoder (and specifically of the same fixed model size).

**Transformer Architectures for Next Token Prediction.** Transformers have become the de facto backbones for next-token prediction tasks, leading to several variants such as encoder-decoder, causal decoder-only, and prefix decoder-only models. In the encoder-decoder model (Lewis et al., 2019; Chung et al., 2024), similar to the vanilla Transformer (Vaswani et al., 2017), the encoder transforms the input tokens into conditioning features, and the decoder auto-regressively predicts the target tokens by using cross-attention over the encoded representation and causal attention over the output tokens. In contrast, the causal decoder-only model (Brown et al., 2020; Chowdhery et al., 2023) uses only the Transformer decoder and applies causal attention to all tokens to perform next-token prediction, ensuring that each token attends only to previous tokens. The prefix decoder-only model (Raffel et al., 2020; Wu et al., 2021) is similar to the causal decoder-only model but differs in that it applies non-causal attention (i.e., full self-attention) to the input sequence (see Figure 8 for visualizations of the attention patterns in these variants).

With the development of these models, recent studies have investigated the performance of each variant across various tasks. Notably, Wang et al. (2022) examined the zero-shot generalization performance of each model along with various objectives, and Ding et al. (2024) analyzed the performance of causal decoder-only and prefix decoder-only models in in-context learning. Patel et al. (2023) showed that bidirectional language models, trained with denoising objectives such as masked language modeling, can be prompted in an auto-regressive manner, achieving performance comparable to larger decoder-only Transformers. Building on this foundation, ENTP explores the training of bidirectional language models using auto-regressive objectives.

## 3 Preliminaries

**Sequence-to-Token Functions and Autoregression.** Given a vocabulary $V$ (which we traditionally think of as some finite set, but could in general be any arbitrary perhaps uncountable set e.g. $\mathbb{R}^d$), we can define a sequence over $V$ as $(x_1, ..., x_n)$ where $x_1, ..., x_n \in V$. Let $V^* = \{(x_1, ..., x_n) : n \in \mathbb{N}; x_i \in V\}$ be the set of all sequences generated by $V$. Then, we say that $f : V^* \to V$ is a sequence-to-token function.

We can view a causal model as a map from an input sequence to an output sequence with the causality constraint that the $i$'th output token depends only on the first $i$ input tokens. Mathematically, we enforce this causality constraint by characterizing our causal model, $\mathcal{T}_f : V^* \to V^*$, with some sequence-to-token function $f$ where on input sequence $(x_1, ..., x_n)$ we have that

$$\mathcal{T}_f(x_1, ..., x_n) \coloneqq (f(x_1), f(x_1, x_2), ..., f(x_1, ..., x_n)). \tag{1}$$

Observe that a sequence-to-token function $f$ can be used auto-regressively to generate tokens from some initial sequence $(x_1, ..., x_n)$ via the following update rule:

$$x_{n+i} \coloneqq f(x_1, ..., x_{n+i-1}). \tag{2}$$

This can also be viewed as a special case of the causal model, where the input sequence is chosen so that

$$\mathcal{T}_f(x_1, x_2, ..., x_n) = (x_2, x_3, ..., x_{n+1}). \tag{3}$$

Hence, if we are trying to learn a causal or auto-regressive model, it suffices to learn the sequence function that generates it. Thus in this paper, we focus on the type of sequence functions that encoders versus decoders can learn and express.

**Encoders and Decoders.** We will use the letters $\mathcal{E}$ and $\mathcal{D}$ respectively to refer to encoders and decoders. In this paper, both models refer to variants of the Transformer architecture introduced in Vaswani et al. (2017), where the only difference lies in the masking used on the attention scores (decoder uses a causal mask while encoder allows full attention, as illustrated in Figure 1).

The model size of a Transformer is determined by two parameters:

- $L$: number of Transformer blocks.

- $D$: embedding dimension.

As Transformers are sequence-to-sequence maps, we will use subscript notation where $\mathcal{T}(x_1, ..., x_n)_i$ denotes the $i$'th value in the output sequence. We also allow our models the option to use positional embeddings. Tilde-notation i.e. $\widetilde{\mathcal{E}}$ and $\widetilde{\mathcal{D}}$ will denote models that do not use positional embeddings. For token embeddings $x_1, ..., x_n$ and positional embeddings $p_1, ..., p_n$:

$$\mathcal{E}(x_1, ..., x_n) \coloneqq \widetilde{\mathcal{E}}(x_1 + p_1, ..., x_n + p_n).$$

In our experiments, we will use encoder and decoder models that have access to trainable positional embeddings.

**Encoder and Decoders as Causal Models.** Given encoder $\mathcal{E}$ and decoder $\mathcal{D}$, we can associate them with sequence-to-token functions as follows:

$$f_{\mathcal{E}} : (x_1, ..., x_n) \mapsto \mathcal{E}(x_1, ..., x_n)_n$$
$$f_{\mathcal{D}} : (x_1, ..., x_n) \mapsto \mathcal{D}(x_1, ..., x_n)_n.$$

We then have $\mathcal{T}_{\mathcal{E}}$ and $\mathcal{T}_{\mathcal{D}}$ as the causal models of $\mathcal{E}$ and $\mathcal{D}$ when used as sequence functions $f_{\mathcal{E}}$ and $f_{\mathcal{D}}$ respectively.[1]

Under this characterization, we make two observations. Firstly, we can view $\mathcal{T}_{\mathcal{E}}$ as an explicit and necessary way to introduce causality to the encoder $\mathcal{E}$ since there is nothing implicit to the encoder that forces causality. Secondly, in juxtaposition to the previous statement, the causal model $\mathcal{T}_{\mathcal{D}}$ is exactly equivalent to just using $\mathcal{D}$, that is $\mathcal{T}_{\mathcal{D}} = \mathcal{D}$. This is because $\mathcal{D}$ enforces causality implicitly via the attention mask (see Appendix A.2 for formal proof). Therefore, the explicit enforcement becomes redundant. To concretely illustrate these two observations we additionally provide a full derivation of the computation in a toy two-layer example comparing difference between encoder and decoder in Appendix A.4.

## 4 Expressive Power of Encoder-only vs. Decoder-only Transformers

Given that we can learn causal functions (defined as causal model in preliminary) using either encoders and decoders, the natural question to ask is how the expressive power of each model is related, i.e. can the encoder express more causal functions than a decoder of the *same* model size? Or perhaps they express the exact same class of causal functions? Towards answering this question, one trivial observation is that one-layer decoders and encoders are equivalent (formal proof in Appendix A.3) which directly implies the existence of causal functions[2] that both architecture can model exactly.

Now, what about causal functions that a decoder can model but not encoder, or vice versa — that encoder can model but not decoder? These functions exist too, as we show in the following two theorems:

**Theorem 4.1.** *For any $L \geq 2$ and $D \geq 1$, there exists a position-free decoder $\widetilde{\mathcal{D}}$ that has $L$-layers and embedding dimension $D$, such that for any encoder $\mathcal{E}$, there exists some input sequence $(x_1, x_2, ...)$ with $x_1, x_2, ... \in \mathbb{R}^D$, and $\mathcal{T}_{\widetilde{\mathcal{D}}}(x_1, x_2, ...) \neq \mathcal{T}_{\mathcal{E}}(x_1, x_2, ...)$.*

**Theorem 4.2.** *For any $L \geq 2$ and $D \geq 1$, there exists a position-free encoder $\widetilde{\mathcal{E}}$ that has $L$-layers and embedding dimension $D$, such that for any decoder $\mathcal{D}$ with positional embeddings satisfying $p_1 \neq p_2$, there exists some input sequence $(x_1, x_2, ...)$ with $x_1, x_2, ... \in \mathbb{R}^D$, and $\mathcal{T}_{\widetilde{\mathcal{E}}}(x_1, x_2, ...) \neq \mathcal{T}_{\mathcal{D}}(x_1, x_2, ...)$.*

The above two theorems are existential in nature. Informally, Theorem 4.1 says that if we consider causal model defined over the entirety of $\mathbb{R}^D$ as its vocabulary, we can find some decoder, for which any encoder will differ from it on some input sequence. Theorem 4.2 makes a similar (albeit weaker statement) in the

---

[1]To be fully consistent with notation in equation 1, we should denote them as $\mathcal{T}_{f_{\mathcal{E}}}$ and $\mathcal{T}_{f_{\mathcal{D}}}$ respectively. However we abuse notation and use $\mathcal{T}_{\mathcal{E}}$ and $\mathcal{T}_{\mathcal{D}}$ for sake of simplicity.

[2]Here causal function refers to sequence-to-sequence function where the outputs are only determined by current and previous inputs, i.e. $\mathcal{F}(x_1, x_2, ..., x_n)_i$ only depends on $x_1, x_2, ..., x_i$, and not $x_j$ for any $j > i$.

other direction; namely the existence of a causal function computable by an encoder, but not by any decoder that uses "non-trivial" positional embeddings (e.g. embeddings for different positions are unique). Detailed proof of both theorems are deferred to Appendix A.

Of course, the setting and assumptions of the above two statements are not necessarily very realistic. For one, they focus on general class of causal models rather than only auto-regressive ones. The assumption of unbounded domain is also not realistic as in practice decoders are trained and used over a finite domain of tokens, each with some fixed embeddings. And specific to Theorem 4.2, no claim is made about decoders that do not use positional embeddings. But despite the limitations, these theorems give an indication that the expressive power of encoder and decoder model are different — despite the almost identical description modulo the attention mask. Changing the mask on the attention scores causes significant changes to the properties of the model. Thus, in the following sections we propose an auto-regressive tasks and run experiments comparing encoders and decoders that corroborates this view.

While the previous theorems show that ENTP and decoder-only Transformers can express different causal functions exactly, they do not address their abilities to approximate functions. In practice, models only need to approximate functions, not represent them exactly. Under this relaxed setting, ENTP can replicate causal masking by augmenting their queries and keys, effectively simulating a decoder.

Given the decoder's queries and keys, $Q = [q_1, q_2, \ldots, q_n]^T$ and $K = [k_1, k_2, \ldots, k_n]^T$, the causal mask can be replicated in a noncausal attention mechanism by augmenting the queries and keys. For a sequence of length $n = 4$, the queries and keys are augmented as follows:

$$
\tilde{Q} = \begin{bmatrix} q_1^T & -\infty & 0 & 0 & 0 \\ q_2^T & 0 & -\infty & 0 & 0 \\ q_3^T & 0 & 0 & -\infty & 0 \\ q_4^T & 0 & 0 & 0 & -\infty \end{bmatrix} \quad \tilde{K} = \begin{bmatrix} k_1^T & 0 & 0 & 0 & 0 \\ k_2^T & 1 & 0 & 0 & 0 \\ k_3^T & 1 & 1 & 0 & 0 \\ k_4^T & 1 & 1 & 1 & 0 \end{bmatrix} \tag{4}
$$

Multiplying these, we obtain:

$$
\tilde{Q}\tilde{K}^T = \begin{bmatrix} q_1^T k_1 & -\infty & -\infty & -\infty \\ q_2^T k_1 & q_2^T k_2 & -\infty & -\infty \\ q_3^T k_1 & q_3^T k_2 & q_3^T k_3 & -\infty \\ q_4^T k_1 & q_4^T k_2 & q_4^T k_3 & q_4^T k_4 \end{bmatrix}, \tag{5}
$$

which is exactly equivalent to applying causal attention. Functionally, $-\infty$ is a sufficiently large negative constant.

For any sequence length $n$, given an orthogonal basis $a_1, a_2, \ldots, a_n$,

$$
\tilde{q}_i = \begin{bmatrix} q_i \\ -\infty a_i \end{bmatrix} \quad \text{and} \quad \tilde{k}_i = \begin{bmatrix} k_i \\ \sum_{j=1}^{i-1} a_i \end{bmatrix},
$$

so that

$$
q_i^T k_j = \begin{cases} q_i^T k_j & \text{if} \quad i \geq j \\ -\infty & \text{if} \quad i < j \end{cases}
$$

Furthermore, from the Johnson-Lindenstrauss lemma, it is possible to approximate these augmented keys and queries using embedding extensions of only $O(\log n)$ dimension, by using a basis that is approximately orthogonal.

Thus, provided that positional information is accurately propagated through the Transformer via skip connections, an encoder can effectively replicate causal attention.

> **Finding 1:** *While ENTP and decoder-only Transformers express distinct sets of causal functions exactly, encoders can approximate any causal function that a decoder can compute.*

As will be shown in Section 6.2, the inverse does not hold: there exist some ENTP models that decoder-only Transformers cannot approximate.

## 5 Time and Space Complexity Comparisons

Inspired by the different computational models of encoder-only and decoder-only Transformers, we characterize the causal sequence functions learnable by encoders and decoders based on their required computational complexity. We give an informal comparison of encoders and decoders in terms of their required time and space complexities — both over the entire sequence and for each additional token. We propose Count3, which is closely related to Match3 (Sanford et al., 2024b), to highlight the "gap" between the complexity of encoders and decoders. Count3 is feasible for an encoder but challenging for a decoder due to its limited computation complexity.

**Time Complexity Comparison.** Decoder-only Transformers, using KV-Cache, take $O(n)$ time to generate each token, and $O(n^2)$ time to generate an entire sequence. Because an ENTP has to compute the entire attention matrix for every token, it takes $O(n^2)$ time to generate each token. Thus, it takes $O(n^3)$ time to generate the entire sequence. While this implies that ENTP is more compute-intensive (i.e., ENTP will be slower than decoder-only Transformer), this also implies that ENTP can express more compute-intensive functions than decoders. Specifically, since the total amount of compute that decoders use for generating n tokens is $O(n^2)$, they cannot run any algorithm whose runtime is $\omega(n^2)$ (strictly greater than quadratic time).

**Space Complexity Comparison.** Both encoder-only and decoder-only use $O(n)$ space complexity to generate an entire sequence. Although the standard implementation of attention uses $O(n^2)$ space, attention can be implemented using only $O(n)$ space. For details of algorithmic implementation of attention using constant space per token, refer to Algorithm 1 in the appendix. Thus, we need a more detailed approach to find a difference between the space complexity of the two models.

Towards this end, we classify memory used for the computation over the current token as either precomputed or additional. Precomputed memory stores values from computation over past tokens. Values stored in precomputed memory *persist*, and are used for computation over current and future tokens, e.g. the keys and values of previous tokens for a decoder. Additional memory stores values that depend on the current token, e.g. keys and values of the current token.

When generating the $n$th token, a decoder uses $O(n)$ precomputed memory to store keys and values of previous tokens and $O(1)$ additional memory to compute results over the current token. An encoder computes everything from scratch for each token, so it uses $O(n)$ additional memory and no precomputed memory. Under this view, there is a space complexity gap between encoder and decoder.

Table 1: Complexity for next-token inference.

| Complexity | Encoder-only | Decoder-only |
|---|---|---|
| Additional Time Complexity | $O(n^2DL)$ | $O(nDL)$ |
| Precomputed Space Complexity | N/A | $O(nDL)$ |
| Additional Space Complexity | $O(nD)$ | $O(D)$ |

Most of our complexity analysis focuses on Transformers with fixed sizes, so we primarily consider complexity with respect to the sequence length $n$. However, we also account for the embedding dimension $D$ and the number of layers $L$ in Table 1. Both encoder-only and decoder-only Transformers use $O(LD)$ time because the attention operation is performed $O(L)$ times, and computing each query, key, and value vector is $O(D)$. In the case of multi-head attention, we assume $D = hd$, where $h$ is the number of heads and $d$ is the dimension of the query, key, and value vectors. A decoder uses $O(nhdL) = O(nDL)$ precomputed space because it stores $nhL$ query, key, and value $d$-dimensional vectors. Both encoders and decoders use $O(D)$ additional space for current token's embedding vector — and we specifically note that there is no dependence on $L$ as

Transformer do computation sequentially on the layer (i.e. all the additional computation required for layer $\ell$ is done before all the additional computation required for layer $\ell + 1$). Thus, we can do the computation over $L$ layers using $O(D)$ space by overwriting computation over previous layers.

## 6 Task-Specific Analysis with Count3

Consider the sequence function that maps an input sequence of positive integers $x_1, x_2, ..., x_n$ to the number of pairs $x_i, x_j$ where the modulo-$n$ sum of $x_i, x_j$ and $x_n$ is equal to 0. More formally,

$$\text{Count3}(x_1, x_2, ..., x_n) := \left| \left\{ (i, j) \in [n]^2 : x_i + x_j + x_n \equiv 0 \pmod{n} \right\} \right| \pmod{n}. \tag{6}$$

Count3 is an augmented version of Match3 (Sanford et al., 2024b). As shown in Sanford et al. (2024b), the triple-wise relationships, used in both Match3 and Count3, are difficult for Transformers to represent, because of the pairwise nature of self-attention.

Note that there exists algorithms that can compute Count3 on some length-$n$ input sequence $x_1, x_2, ..., x_n$ in either $O(n^2)$ time and $O(1)$ space or in $O(n)$ time and $O(n)$ space. See Algorithm 2 and Algorithm 3 in Appendix E for exact pseudocode implementations. In brief, Algorithm 2 iterates through all $n^2$ pairs checking if they meet the modulo-$n$ sum requirement. Algorithm 3 uses the fact that $x_i + x_j + x_n \equiv 0 \pmod{n}$ is equivalent to $x_i + x_n \equiv -x_j \pmod{n}$. In two linear passes, it counts each $-x_j \pmod{n}$ and stores each count in a table, then sums the values in the table for each $x_i + x_n \pmod{n}$. Now, given these two algorithms, we make the following conjecture:

**Conjecture 6.1.** *Given an algorithm $\mathcal{A}$ that computes $\text{Count3}(x_1, x_2, ..., x_n)$, at least one of the following must hold true:*

*(i) $\mathcal{A}$ requires $\Omega(n^2)$ time with access to $x_n$*

*(ii) $\mathcal{A}$ requires $\Omega(n)$ space storing values unique to $n$.*

This conjecture seems plausible given that both Algorithm 2 and Algorithm 3, which we consider to be optimal, adhere to it. Algorithm 2 uses $O(n^2)$ time after accessing $x_n$ (line 4 of Algorithm 2) and Algorithm 3 requires $O(n)$ memory, where the stored values are a function of $n$ (line 4 of Algorithm 3).

### 6.1 Expressivity of Transformers on Count3

We analyze the expressivity of decoder-only Transformers and ENTP on Count3. From Conjecture 6.1, we have the following lemma:

**Lemma 6.2.** *Given that Conjecture 6.1 holds and assuming $O(\log n)$ precision, any decoder-only Transformer with fixed embedding dimension $D$ satisfying $\mathcal{D}(x_1, ..., x_m)_m = \text{Count3}(x_1, ..., x_m)$ for all sequences of length $m \leq n$ must have $L = \Omega(n)$.* [3]

*Proof.* Let $\mathcal{D}$ be a decoder with $L$ layers and embedding dimension $D$ satisfying $\mathcal{D}(x_1, ..., x_m)_m = \text{Count3}(x_1, ..., x_m)$ for all sequences of length $m \leq n$. We can use $\mathcal{D}$ as an algorithm to compute $\text{Count3}(x_1, ..., x_n)$, by outputting $\mathcal{D}(x_1, ..., x_n)_n$ on input $(x_1, ..., x_n)$. Thus either condition (i) or (ii) of Conjecture 6.1 must hold when $\mathcal{D}$ computes the output sequence over $(x_1, ..., x_n)$.

**Case 1: (i) is true.** In this case $\mathcal{D}$ requires $\Omega(n^2)$ time with access to $x_n$, i.e. $\mathcal{D}$ uses $\Omega(n^2)$ time for the last token. From Table 1, we know that $\mathcal{D}$ uses $O(nLD)$ time for each token. Thus condition (i) is true only if we have that $nLD = \Omega(n^2)$. Since $D$ is fixed, it follows that $L = \Omega(n)$.

---

[3] $O(\log n)$ precision is not required for Lemma 6.2, but it is included to be consistent with Lemma 6.3.

Figure 2: An example of a sequence used in a Count3 experiment.

**Case 2: (ii) is true.** In this case, $\mathcal{D}$ requires $\Omega(n)$ space storing values unique to $n$. Because decoders are causal, we have $\mathcal{D}(x_1, ..., x_n)_i = \text{Count3}(x_1, ..., x_i)$ for all $i \in [n]$. Then since we assume (ii) is true, computing $\mathcal{D}(x_1, ..., x_n)_i$ requires $\Omega(i)$ space for each $i \in [n]$. Furthermore by the uniqueness assumption of (ii), for $i \neq j$, the values stored when computing $\text{Count3}(x_1, ..., x_i)$ are different from the values stored when computing $\text{Count3}(x_1, ..., x_j)$. Since decoders are causal, the space used to compute $\mathcal{D}(x_1, ..., x_n)_i$ cannot be overwritten when computing $\mathcal{D}(x_1, ..., x_n)_j$, for $j > i$. Hence, when $\mathcal{D}$ computes $\text{Count3}(x_1, ..., x_n)$, it uses $\Omega(i)$ space to compute $\text{Count3}(x_1, ..., x_i)$, for each $i \in [n]$. Thus, $\mathcal{D}$ uses $\Omega\left(\sum_{i \in [n]} i\right) = \Omega(n^2)$ space to compute $\text{Count3}(x_1, ..., x_n)$. From Table 1, we know that $\mathcal{D}$ uses $O(nLD)$ space for the entire sequence. Then $nLD = \Omega(n^2)$. Since $D$ is fixed, it follows that $L = \Omega(n)$.

Finally, as $L = \Omega(n)$ is a necessary condition for both conditions (i) and (ii), Theorem 6.2 follows. $\square$

**Lemma 6.3.** *Assuming $O(\log n)$ precision, there exists an encoder $\mathcal{E}$ with $L = O(1)$ and $D = O(1)$ such that $\mathcal{E}(x_1, ..., x_m)_m = \text{Count3}(x_1, ..., x_m)$ for all sequences of length $m \leq n$.* [4]

Proof of Lemma 6.3 is in Appendix A.7.

*Remark* 6.4. With linear chain-of-thought (generating $O(n)$ tokens before answering), a decoder would be able to perform Count3. We provide a RASP[5] (Weiss et al., 2021) program, Algorithm 6, to demonstrate this.

### 6.2 Implications of Lemma 6.2 for Decoder Approximation of ENTP

Lemma 6.2 addresses the exact computation of Count3. Many continuous Transformer realizations can interpolate such a function. This is in contrast to Section 4, which focuses on exact replication and approximation of continuous functions mapping sequences of vectors in $\mathbb{R}^d$.

If no decoder-based language model can exactly replicate a given ENTP-based language model (i.e., produce outputs whose argmax predictions match), then there must also be a limit on how closely a decoder can approximate its output embeddings, assuming there is a nontrivial gap between the largest and second largest logit values (as observed for our ENTP models on the Count3 task, which achieve near-zero loss in Figure 3). Thus, combining Lemma 6.2 with the observation above, a decoder cannot approximate arbitrarily well an ENTP model that successfully solves Count3, without its size scaling with sequence length.

### 6.3 Count3 **Experiments**

We train small decoder-only and encoder-only Transformers on auto-regressive sequences generated from Equation (6). To generate unique sequences, we start each sequence with a seed containing 16 random integers between 0 and 63. Then we extend the sequence to 64 integers using Equation (6) (see Figure 2). Seeds are generated randomly during training and evaluation. With $64^{16} \approx 1.16 \times 10^{77}$ possible seeds, the chance of significant duplication among the $1.28 \times 10^7$ seeds used for training and evaluation is negligible.

---

[4]We assume $x_1, ..., x_m < m$.

[5]RASP (Weiss et al., 2021) is a programming language that describes Transformer computations, by mapping attention and feed-forward computation into simple primitives.

Consequently, the reported results effectively correspond to "test" loss and accuracy. The seed portion of the sequence was not used to compute the loss during training, so the model was only trained on the deterministic part of the sequence.

As shown in Figure 3, the decoder-only Transformers demonstrate some ability to learn patterns related to the distribution of numbers in Count3 sequences, but they completely fail to learn the task. In contrast, the encoder successfully learns the task with near-perfect accuracy.

We also evaluated Prefix decoder-only models (Raffel et al., 2020; Wu et al., 2021), which perform non-causal attention for the prefix portion of the sequence. We used the same experimental setup, and set the 16-integer seed as the prefix. As shown in Figure 3, while the Prefix decoder-only model slightly outperforms the decoder-only model, it also fails to learn the triplet counting task. This suggests that incorporating full attention over parts of the sequence in a decoder-only model is insufficient for solving tasks with Count3-level complexity.

To demonstrate that ENTP is effective for larger pre-trained models, we fine-tuned BERT (Devlin et al., 2019)[6] using the ENTP approach under the same experimental conditions. As shown in Figure 3, BERT combined with ENTP successfully learned triplet counting. Notably, as BERT is pre-trained and larger compared to the medium transformer, it converged more quickly and achieves higher accuracy.

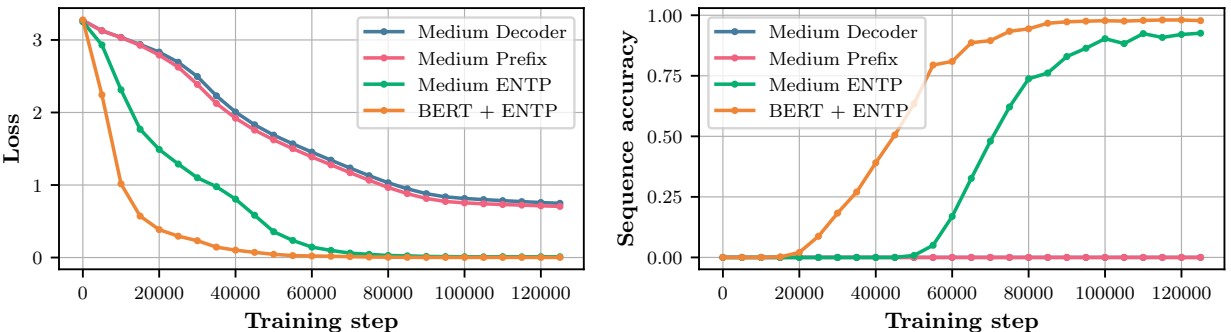

Figure 3: **Training loss (left) and sequence accuracy curve (right) for the** Count3**.** ENTP successfully learns to perform the Count3 task, but the decoder-only Transformers and prefix Transformers struggle to learn it.

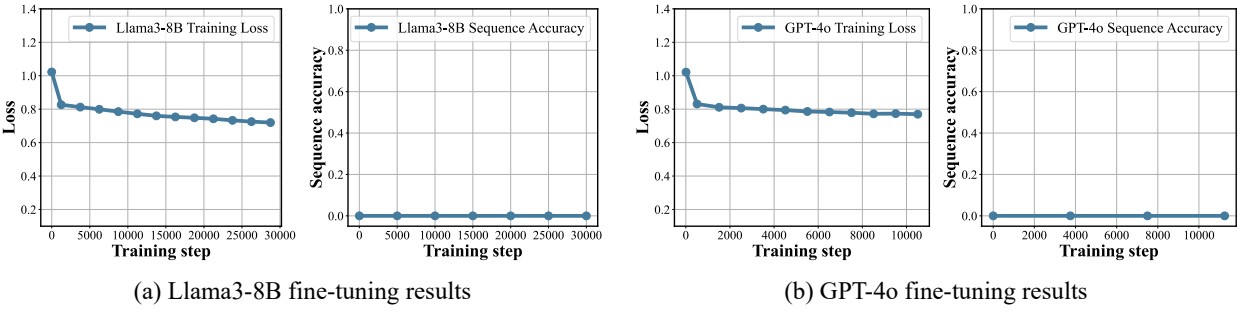

(a) Llama3-8B fine-tuning results          (b) GPT-4o fine-tuning results

Figure 4: **Results of LLM fine-tuning on** Count3**.** Decoder-only Transformer, even at large scales, fail to efficiently learn the Count3 task.

To investigate the performance of decoder-only large language models (LLMs) on the Count3 task, we fine-tune Llama-3 (Dubey et al., 2024) and GPT-4o (Achiam et al., 2023) using sequences of 64 integers introduced previously. To enable the LLMs to leverage their knowledge, we also include the code for Algorithm 2 in the

---

[6]BERT is an encoder-only Transformer trained using the masked language modeling objective.

prompt, asking the models to provide the result after executing the code (see Table 5 for the full prompt). As shown in Figure 4, the LLMs struggle with the Count3 task, which is consistent with our small-scale experiment. It demonstrates that the suggested characteristics of causal decoder-only models hold true even at large scales. We provide the validation of the prompt design used, as well as details about the LLM fine-tuning, in the Appendix C.1.

Finally, we remark that Count3 sequences have low Kolmogorov complexity, meaning the smallest program generating them is short—upper bounded by the six-line Algorithm 2. However, as shown in Theorem 6.2 and the experiments above, no decoder-only Transformer, regardless of size, can efficiently learn the task.

> **Finding 2:** *ENTP can easily learn* Count3*, but large decoder-only Transformers cannot.*

### 6.4   Similar Function Learnable by Decoder

Motivated by the question of how we need to change Count3 so that it can be learned by a decoder, we examine a modified version of Match3 (Sanford et al., 2024b).

$$\text{Match3}'(x_1, x_2, ..., x_n) := \begin{cases} 1 & \exists\,(i, j) : x_1 + x_i + x_j = 0 \pmod{128} \\ 0 & \text{otherwise} \end{cases}$$

There are several key differences between Count3 and Match3$'$: (1) Match3$'$ uses a fixed modulus, whereas Count3 employs the sequence length as the modulus. This simplifies the decoder's task, as the modulus remains constant across all tokens, enabling reuse of intermediate values from previous tokens; (2) Match3$'$ operates on triplets $(x_1, x_i, x_j)$ rather than $(x_i, x_j, x_n)$. By using $x_1$ instead of $x_n$, it becomes easier for the decoder since $x_1$ remains unchanged for different tokens, facilitating the reuse of intermediate values across tokens; (3) Match3$'$ checks for the existence of a condition rather than counting occurrences. Counting is challenging to implement within the attention mechanism without scaling values by the sequence length. Due to the causal mask, scaling value vectors by sequence length is not straightforward.

We provide RASP (Weiss et al., 2021) program, Algorithm 5 that satisfies $\mathcal{D}(x_1, x_2, ..., x_n)_n = \text{Match3}'(x_1, x_2, ..., x_n)$ for sequences of any length, assuming $O(\log n)$ precision. We train small Transformers to verify that both decoders and encoders can perform Match3$'$, and find that both models can perform Match3$'$ with high accuracy.

Table 2: Match3$'$ performance.

| Model | Min Loss | Token Accuracy | Full Sequence Accuracy |
|---|---|---|---|
| Medium Decoder (6 layer) | **0.0001** | **99.99**% | **99.92**% |
| Medium Encoder (6 layer) | 0.0016 | 99.97% | 99.50% |

## 7   Experimental Results for Small-Scale Language Modeling Tasks

Scaling up ENTP to large models remains challenging, but it can still be tested effectively at limited scales on simple tasks. To assess its capabilities, we train ENTP models on small yet representative language modeling benchmarks. These tasks include arithmetic reasoning (Lee et al., 2023; McLeish et al., 2024; Zhou et al., 2024), in-context learning with synthetic data (Garg et al., 2022; Bai et al., 2023; Ding et al., 2024), and language modeling on small datasets (Polo et al., 2024; Sinha et al., 2019). By focusing on these controlled experiments, we can evaluate ENTP's potential while keeping computational demands manageable.

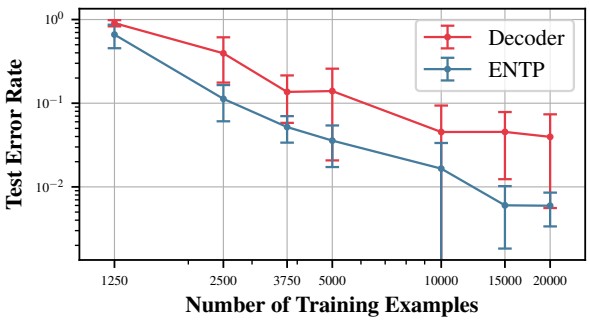
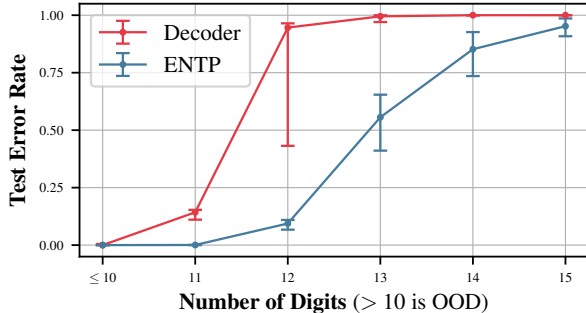

Figure 5: **Addition Sample Complexity.** The train and test datasets include numbers with up to 3 digits.

Figure 6: **Addition Length Generalization.** The train set has numbers up to 10 digits, and the test set has up to 15.

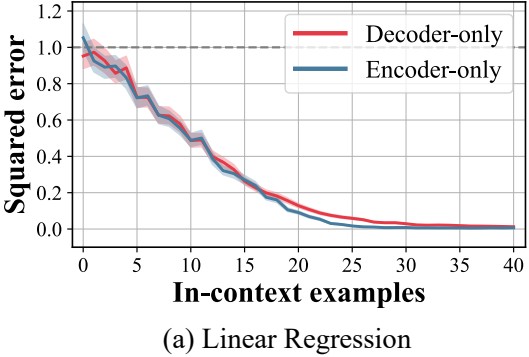
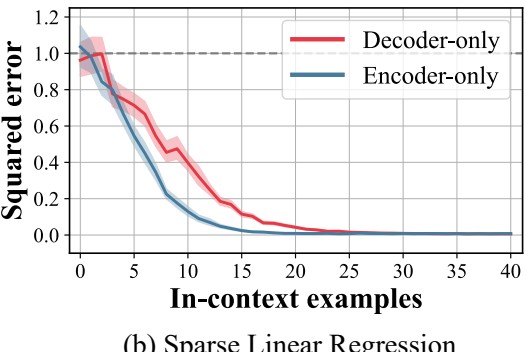

(a) Linear Regression

(b) Sparse Linear Regression

Figure 7: **Results of in-context learning experiment.** The encoder-only models demonstrate superior performance across both function classes compared to the decoder-only models.

## 7.1 Generalization on Addition

We test the sample complexity and length generalization capabilities of decoders and encoders using addition tasks. We use the reversed addition format (`$123+456=975$`) from Lee et al. (2023). We find that encoders exhibit lower sample complexity compared to decoders, as seen in Figure 5, meaning they require fewer training examples to achieve similar performance. Additionally, encoders demonstrate superior ability to generalize to longer sequences, as shown in Figure 6. We provide more experimental details and results in Appendix C.3.

> **Finding 3:** *ENTP achieves superior generalization compared to decoder-only Transformers on both in-distribution and out-of-distribution data.*

## 7.2 In-Context Learning

We consider the problem of learning a function class $\mathcal{F}$ using in-context example (Garg et al., 2022). In this problem, a function $f \in \mathcal{F}$ is sampled from a distribution $\mathcal{D}_\mathcal{F}$, and a sequence of random inputs is sampled i.i.d. from $\mathcal{D}_\mathcal{X}$, forming a prompt $P : (x_1, f(x_1), x_2, f(x_2), ..., x_N, f(x_N))$. The objective is for the model to in-context learn the function $f$ from the prompt $P$ and predict $f(x_\text{query})$ for a new input $x_\text{query}$.

We examine two types of function classes: linear function and sparse linear function. For both function classes, we sample $x_i$ from a Gaussian distribution and utilize the squared error as loss function.

We present the in-context learning results according to the number of in-context examples for each function class in Figure 7. ENTP demonstrates superior performance compared to the decoder-only models in in-context learning for both function classes. A detailed description of each function class, along with additional experimental results for the non-linear function classes, is provided in the Appendix C.2.

**Finding 4:** *For in-context learning, ENTP shows competitive performance compared to decoders.*

### 7.3 Natural Language Tasks

We train Transformer models on the OpenWebText dataset (Gokaslan et al., 2019), an open-source replication of WebText used for GPT-2 training (Radford et al., 2019), using a next-token prediction objective. We conducted OpenWebText training of ENTP models with two different seeds. We use medium models, described in Table 6, and hyperparameters from Table 8. As shown in Table 3, the encoder-only Transformer slightly outperforms the decoder-only Transformer.

To assess commonsense reasoning, we use the TinyWinoGrande benchmark (Polo et al., 2024), which tests pronoun resolution. After pretraining on OpenWebText, both models undergo zero-shot evaluation on this benchmark.

We further evaluate the models on an NLP classification task using CLUTRR (Sinha et al., 2019), which requires identifying familial relationships from text. After fine-tuning on CLUTRR, both models are tested on a holdout set. We randomly select 10,000 examples for the training set and conduct each experiment using three different random seeds.

As shown in Table 4, ENTP achieves higher accuracy than the decoder-only model on both the TinyWino-Grande and CLUTRR tasks, demonstrating its potential for diverse natural language tasks.

Table 3: Minimum values of training and validation loss, as well as perplexity, for decoder-only and encoder-only Transformers on the OpenWebText dataset. Results are averaged over two different random seeds. We report results for the individual training runs in Table 7

| Model | Train Loss | Validation Loss | Train Perplexity | Validation Perplexity |
|---|---|---|---|---|
| Decoder-only | $4.689 \pm 0.006$ | $4.701 \pm 0.005$ | $108.7 \pm 0.603$ | $110.0 \pm 0.496$ |
| Encoder-only | $\mathbf{4.636 \pm 0.008}$ | $\mathbf{4.643 \pm 0.008}$ | $\mathbf{103.1 \pm 0.795}$ | $\mathbf{103.8 \pm 0.786}$ |

Table 4: The performance of decoder-only model and ENTP on the TinyWinoGrande and CLUTRR benchmarks.

| Benchmark | Decoder-only | Encoder-only (ENTP) |
|---|---|---|
| TinyWinoGrande Error Rate | $43.5 \pm 0.5$ % | $\mathbf{38.0 \pm 1.0}$ % |
| CLUTRR Error Rate | $0.9 \pm 0.12$ % | $\mathbf{0.5 \pm 0.03}$ % |

**Finding 5:** *ENTP performs better than decoder-only models on next-token prediction based language modeling, leading to superior performance on downstream natural language tasks.*

## 8 Discussion

In this work, we present theoretical and novel experimental results suggesting that, assuming compute is unlimited, decoder-only Transformers are not the ideal model for sequence modeling. We show that ENTP is more expressive without compromising generalization. Using Theorem 4.1 and 4.2, we find that the classes

of functions encoder-only and decoder-only Transformers can exactly learn are different. We introduce the Count3 task and demonstrate, both theoretically and experimentally, that while ENTP can perform it easily, decoders cannot. We also find that encoders outperform decoders on various auto-regressive tasks, including length generalization and in-context learning.

Despite its advantages, ENTP is currently computationally inefficient for practical deployment. However, it is a valuable architecture for further study, as it can provide deeper insights into the fundamental workings of Transformers. By understanding why ENTP exhibits superior expressiveness and generalization, we may be able to leverage these insights to develop improved next-generation architectures for language models.

A promising future direction is developing a compute-efficient ENTP variant that retains its strengths while narrowing the efficiency gap with decoders. Achieving this balance could lead to models that surpass existing architectures in both performance and scalability, making it a valuable area for further research.

### Acknowledgments

The work of Kangwook Lee is supported in part by NSF CAREER Award CCF-2339978, Amazon Research Award, and a grant from FuriosaAI. Daewon Chae was supported by Hyundai Motor Chung Mong-Koo Foundation.

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

# A  Notation and Deferred Proofs

## A.1  Notation

- $\mathcal{E}, \mathcal{D}$: Encoder/Decoder with positional embeddings.

- $\widetilde{\mathcal{E}}, \widetilde{\mathcal{D}}$: Encoder/Decoder without positional embeddings (with same model weights as $\mathcal{E}, \mathcal{D}$) (which we call position free).

- $L$: number of Transformer blocks.

- $D$: embedding dimension.

- $p_i$: The $i$'th positional embedding where $p_i \in \mathbb{R}^D$.

- $\mathbf{W}^i_{Q/K/V}$: The weight matrix of the **Q**uery, **K**ey and **V**alue respectively of the $i$'th attention block.

- $\mathbf{I}$: The identity matrix.

- $\mathbf{0}$: The zero matrix.

- $(x_1, x_2, x_3, \dots)$: A sequence of values/tokens.

- $\mathcal{T}_{\mathcal{E}}(x_1, x_2, x_3, \dots) := (\mathcal{E}(x_1)_1, \mathcal{E}(x_1, x_2)_2, \mathcal{E}(x_1, x_2, x_3)_3, \dots)$.

For convention, we will use bold capital letters $\mathbf{X}$ to denote matrices, and unbold lowercase letters $x$ to denote vectors.

For theorem 4.1 and theorem 4.2, our model of Transformers will use single-head attention and assume the dimension of the Query and Key are equal to the embedding dimension $D$. We also omit layer normalization and scaling of attention scores by $1/\sqrt{D}$.

## A.2  $\mathcal{T}_{\mathcal{D}} = \mathcal{D}$

*Proof.* The main observation is that the the MLP layers of a decoder are element-wise, and the attention layers of a decoder are causal (i.e. the contextual embedding of the $i$'th token is computed only using the tokens $j \leq i$). We thus have that

$$\mathcal{D}(x_1, \dots, x_i, \dots, x_n)_i = \mathcal{D}(x_1, \dots, x_i)_i$$

for any $i \in [n]$. As a result,

$$
\begin{aligned}
\mathcal{D}(x_1, \dots, x_i, \dots, x_n)_i &= \mathcal{D}(x_1, \dots, x_i)_i \\
&= f_{\mathcal{D}}(x_1, \dots, x_i) \\
&= \mathcal{T}_{\mathcal{D}}(x_1, \dots, x_i, \dots, x_n)_i,
\end{aligned}
$$

for all $i \in [n]$, i.e. $\mathcal{D}(x_1, \dots, x_n) = \mathcal{T}_{\mathcal{D}}(x_1, \dots, x_n)$. $\qquad\square$

## A.3  One-layer Encoder and Decoder are Equivalent

*Proof.* Let $\mathcal{D}$ and $\mathcal{E}$ have the same parameters. Then the query key, and value vectors for the attention layer (denoted $q_i, k_i, v_i$ for each $x_i$ respectively) will be the same for both models.

For one-layer decoder: $\mathcal{D}(x_1, ..., x_n)_i = \text{MLP}\left(\sum_{j=1}^{i} \text{Softmax}(q_i, k_j)v_j\right)$.

For one-layer encoder: $\mathcal{E}(x_1, ..., x_n)_i = \text{MLP}\left(\sum_{j=1}^{n} \text{Softmax}(q_i, k_j)v_j\right)$.

If $i = n$, then $\mathcal{D}(x_1, ..., x_n)_i = \mathcal{E}(x_1, ..., x_n)_i$. Thus, $\mathcal{T}_{\mathcal{D}}(x_1, x_2, \dots, x_n) = \mathcal{T}_{\mathcal{E}}(x_1, x_2, \dots, x_n)$. $\qquad\square$

### A.4 Illustrative Example of Computation over Two Layer Encoder and Decoder

**Setup.** We compare a *two-layer* Transformer *with single head self-attention blocks only*—no MLPs, no residual connections, no layer norms, and no output projections—on a length-2 input $(x_1, x_2)$. For token $i$ attending to token $j$ in layer $\ell$, define the score

$$s_{i \to j}^{(\ell)} := \langle W_Q^{(\ell)} x_i^{(\ell-1)}, W_K^{(\ell)} x_j^{(\ell-1)} \rangle.$$

Here $x_i^{(0)} := x_i$ are inputs, and $x_i^{(\ell)}$ are the outputs of layer $\ell$. For an *encoder*, each token may attend to both positions (1 and 2) in every layer. For a *decoder*, token 1 may attend only to itself, while token 2 may attend to both positions (1 and 2) in *every* layer.

**Layer computations.** Each layer applies only attention with values $W_V^{(\ell)}$. Below we write the length-2 updates explicitly (i.e., the computation when we are trying to generate the output corresponding to the second token in our causal model), expanding the softmax weights directly.

**Decoder**:

$$x_1^{(1)} = W_V^{(1)} x_1,$$

$$x_2^{(1)} = \frac{e^{s_{2\to1}^{(1)}} W_V^{(1)} x_1 + e^{s_{2\to2}^{(1)}} W_V^{(1)} x_2}{e^{s_{2\to1}^{(1)}} + e^{s_{2\to2}^{(1)}}},$$

$$x_1^{(2)} = W_V^{(2)} x_1^{(1)} = W_V^{(2)} W_V^{(1)} x_1,$$

$$x_2^{(2)} = \frac{e^{s_{2\to1}^{(2)}} W_V^{(2)} x_1^{(1)} + e^{s_{2\to2}^{(2)}} W_V^{(2)} x_2^{(1)}}{e^{s_{2\to1}^{(2)}} + e^{s_{2\to2}^{(2)}}}.$$

**Encoder**:

$$x_1^{(1)} = \frac{e^{s_{1\to1}^{(1)}} W_V^{(1)} x_1 + e^{s_{1\to2}^{(1)}} W_V^{(1)} x_2}{e^{s_{1\to1}^{(1)}} + e^{s_{1\to2}^{(1)}}},$$

$$x_2^{(1)} = \frac{e^{s_{2\to1}^{(1)}} W_V^{(1)} x_1 + e^{s_{2\to2}^{(1)}} W_V^{(1)} x_2}{e^{s_{2\to1}^{(1)}} + e^{s_{2\to2}^{(1)}}},$$

$$x_1^{(2)} = \frac{e^{s_{1\to1}^{(2)}} W_V^{(2)} x_1^{(1)} + e^{s_{1\to2}^{(2)}} W_V^{(2)} x_2^{(1)}}{e^{s_{1\to1}^{(2)}} + e^{s_{1\to2}^{(2)}}},$$

$$x_2^{(2)} = \frac{e^{s_{2\to1}^{(2)}} W_V^{(2)} x_1^{(1)} + e^{s_{2\to2}^{(2)}} W_V^{(2)} x_2^{(1)}}{e^{s_{2\to1}^{(2)}} + e^{s_{2\to2}^{(2)}}}.$$

**Takeaway.** In the decoder, causality is *implicit*: $x_1^{(\ell)}$ depends only on $x_1$ for all $\ell$, so previously computed keys/values for the first position can be *reused* when producing the output corresponding to position two in later layers. In the encoder, $x_1^{(1)}$ depends on both $x_1$ and $x_2$ when used to compute the output corresponding to the second token. In other words, we *explicitly* enforces causality by recomputing each of the earlier hidden representations when generating a new token.

### A.5 Proof of theorem 4.1

*Proof.* We first provide a construction for $\widetilde{\mathcal{D}}$. For the attention-block of the first two layers, we use the same weight matrices. Namely, we set $\mathbf{W}_K^1 = \mathbf{W}_K^2 = \mathbf{W}_Q^1 = \mathbf{W}_Q^2 = \mathbf{0}$ and $\mathbf{W}_V^1 = \mathbf{W}_V^2 = \mathbf{I}$. For every other attention block, we set them to the constant zero function by setting $\mathbf{W}_V^i = \mathbf{0}$ for $i \geq 3$. We similarly set the weights and biases of every MLP block to zero. Thus, in essence $\widetilde{\mathcal{D}}$ is just two duplicate attention blocks stacked on top of each other with a skip connection after each attention block.

Now consider three arbitrary vectors $x_1, x_2, x_3 \in \mathbb{R}^D$ and its corresponding sequence $(x_1, x_2, x_3)$. Let us first compute the output of $\widetilde{\mathcal{D}}$ on $(x_1, x_2, x_3)$. The first attention block and skip connection will map the input sequence to the sequence

$$\left( 2x_1, x_2 + \frac{x_1 + x_2}{2}, x_3 + \frac{x_1 + x_2 + x_3}{3} \right).$$

The second attention block and skip connection will then map to the following:

$$\left( 4x_1, \frac{7x_1 + 9x_2}{4}, \frac{23x_1 + 17x_2 + 32x_3}{18} \right).$$

Our first observation from this mapping is that there clearly exists $\tilde{x}_1, \tilde{x}_2, \tilde{x}_3 \in \mathbb{R}^D$ such that $\widetilde{\mathcal{D}}(\tilde{x}_1, \tilde{x}_2, \tilde{x}_3)_3 \neq \widetilde{\mathcal{D}}(\tilde{x}_2, \tilde{x}_1, \tilde{x}_3)_3$. Our second observation is that

$$4x_1 = \frac{7x_1 + 9x_2}{4} \quad \Longleftrightarrow \quad x_1 = x_2,$$

which follows from simplifying the left hand equation.

Now, let us for sake of contradiction assume the existence of some encoder $\mathcal{E}$ such that $\mathcal{T}_{\mathcal{E}}$ exactly replicates $\widetilde{\mathcal{D}}$ on every input sequence. We first claim that the first two positional embeddings $p_1, p_2$ of $\mathcal{E}$ must differ. This follows by our first observation and thus the requirement that $\mathcal{E}(\tilde{x}_1, \tilde{x}_2, \tilde{x}_3)_3 \neq \mathcal{E}(\tilde{x}_2, \tilde{x}_1, \tilde{x}_3)_3$ — which can only happen if $p_1 \neq p_2$ due to the permutation invariance of encoders when there are no positional embeddings. Now as $p_1 \neq p_2$, there exists vectors $y_1, y_2, c \in \mathbb{R}^D$ such that $y_1 \neq y_2$ and $y_1 + p_1 = y_2 + p_2 = c$. It follows immediately that

$$\begin{aligned} \mathcal{E}(y_1)_1 &= \widetilde{\mathcal{E}}(y_1 + p_1)_1 \\ &= \widetilde{\mathcal{E}}(c)_1 \\ &= \widetilde{\mathcal{E}}(c, c)_2 \\ &= \widetilde{\mathcal{E}}(y_1 + p_1, y_2 + p_2)_2 \\ &= \mathcal{E}(y_1, y_2)_2. \end{aligned}$$

But since $y_1 \neq y_2$, by the second observation we made, it must be that $\widetilde{\mathcal{D}}(y_1)_1 \neq \widetilde{\mathcal{D}}(y_1, y_2)_2$. Since we assumed that $\mathcal{T}_{\mathcal{E}}$ exactly replicates $\widetilde{\mathcal{D}}$ on every input sequence, it thus follows that $\mathcal{E}(y_1)_1 \neq \mathcal{E}(y_1, y_2)_2$ — a contradiction. Hence, no such encoder $\mathcal{E}$ exists, which directly implies that we can always find some sequence $(x_1, x_2, \dots)$ where $\widetilde{\mathcal{D}}(x_1, x_2, \dots) \neq \mathcal{T}_{\mathcal{E}}(x_1, x_2, \dots)$. $\square$

## A.6 Proof of theorem 4.2

*Proof.* We first provide a construction for $\widetilde{\mathcal{E}}$. For the attention-block of the first two layers, we use the same weight matrices. Namely, we set $\mathbf{W}_K^1 = \mathbf{W}_K^2 = \mathbf{W}_Q^1 = \mathbf{W}_Q^2 = \mathbf{W}_V^1 = \mathbf{W}_V^2 = \mathbf{I}$. For every other attention block, we set them to the constant zero function by setting $\mathbf{W}_V^i = \mathbf{0}$ for $i \geq 3$. We similarly set the weights and biases of every MLP block to zero. Thus, in essence $\widetilde{\mathcal{E}}$ is just two duplicate attention blocks stacked on top of each other with a skip connection after each attention block.

Now consider two arbitrary vectors $x_1, x_2 \in \mathbb{R}^D$ and its corresponding sequence $(x_1, x_2)$. A brief inspection will reveal that

$$\mathcal{T}_{\widetilde{\mathcal{E}}}(x_1, x_2) = (4x_1, \alpha x_1 + \beta x_2), \tag{7}$$

where $\alpha, \beta > 0$.

Next, we assume the existence of some $\mathcal{D}$ where $p_1 \neq p_2$ and exactly replicates $\widetilde{\mathcal{E}}$. We fix $x_1 = 0$ and let $x_2 = p_1 - p_2$. Observe that $x_2 \neq 0$ as $p_1 \neq p_2$. It follows that there is some constant vector $c \in \mathbb{R}^D$ where

$$\begin{aligned} \mathcal{D}(x_1, x_2) &= \widetilde{\mathcal{D}}(x_1 + p_1, x_2 + p_2) \\ &= \widetilde{\mathcal{D}}(p_1, p_1) \\ &= (c, c). \end{aligned}$$

Now from equation 7, we have that $\mathcal{T}_{\widetilde{\mathcal{E}}}(x_1, x_2) = (0, \beta x_2)$ for some $\beta > 0$. As $x_2 \neq 0$, it follows that $\beta x_2 \neq 0$ and hence $(0, \beta x_2)$ is not a constant sequence — contradicting the output of the decoder. Hence, no $\mathcal{D}$ where $p_1 \neq p_2$ can exactly replicate $\widetilde{\mathcal{E}}$. $\qquad\square$

### A.7 Proof of theorem 6.3

*Proof.* Lemma 6.3 follows from Algorithm 4. From Weiss et al. (2021), a RASP program can be compiled to a Transformer, with a fixed number of Transformer-layers and attention heads. However, it assumes that the MLPs can perform any element-wise operation. Thus, it suffices to show that each MLP needs $O(1)$ layers with respect to sequence length $n$.

We first observe that the largest internal value possible within Algorithm 4 is $n^2$, so there are $n^2 + 1$ distinct internal values including 0. Using $\lfloor 2\log_2 n \rfloor + 1$ bits, we can represent all of these values as unsigned integers. Using floating-point numbers we would also need $\Theta(\log n)$ bits.

All linear element-wise operations in Algorithm 4 can be implemented trivially, since an MLP can perform arbitrary linear transformations. Therefore, we focus on the only nonlinear element-wise function $g : [2n - 1] \times [n] \to [n-1]$:[7]

$$g(a,b) = \begin{cases} a, & a < b \\ a - b & a \geq b. \end{cases}$$

Using a constant number of linear operations and ReLU functions, $g$ can be constructed as follows:

$$g(a,b) = \mathrm{ReLU}(a - M\,\mathrm{ReLU}(a - b + \epsilon)) + \mathrm{ReLU}(a - b),$$

where $0 < \epsilon < 1$ and $M \geq \frac{2n}{\epsilon}$.

Because $g$ can be implemented using a constant number of linear operations and ReLU functions, it can be implemented by an MLP with ReLU activation functions and $O(1)$ depth.

Because of skip-connections, we can concatenate MLPs by zeroing out attention. Then we can create any MLP of $O(1)$ depth.

Thus, it is possible to construct an encoder $\mathcal{E}$ with $L = O(1)$ and $D = O(1)$ such that $\mathcal{E}(x_1, \ldots, x_m)_m = f_{\mathrm{TC}}(x_1, \ldots, x_m)$ for all sequences of length $m \leq n$. $\qquad\square$

## B Attention patterns of different Transformer architectures

In Figure 8, we provide the visualization of attention patterns of encoder-only, decoder-only, prefix decoder-only, and encoder-only models.

## C Experiment Details and Additional Results

In all experiments, we train encoders in the same manner as decoders, processing entire sequences in each batch with a single gradient optimization step. Although this approach does not offer the same efficiency benefits for encoders as it does for decoders, we adopt it to maintain consistency between the training processes of both models.

### C.1 Count3 **with LLM**

In the main paper, we fine-tuned two LLMs for Count3: Llama3-8B (Dubey et al., 2024) and GPT-4o (Achiam et al., 2023). Here, we provide the details about the fine-tuning of each model. For GPT-4o, we used the official API, setting the batch size to 4 and the learning rate multiplier to 10. For Llama3-8B, we employed LoRA fine-tuning Hu et al. (2022) with a batch size of 16 and a learning rate of $1.4 \times 10^{-4}$. Regarding

---

[7]This function implements modular division for a bounded range. $g(a,b) = a \mod b$, when $0 \leq a < 2b$.

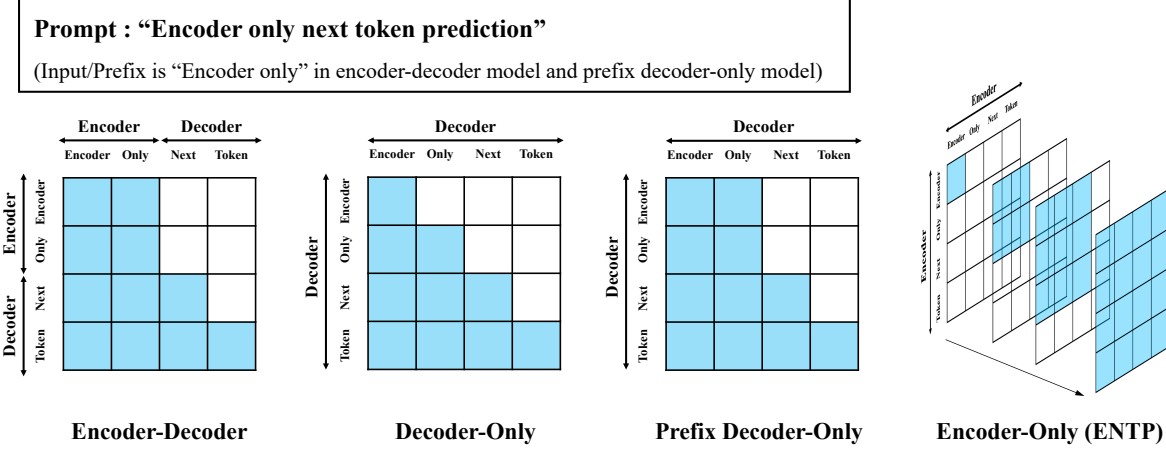

Figure 8: **Attention patterns of different Transformer architectures in next token prediction.** Encoder-decoder and prefix decoder-only models first perform full attention on the input (prefix), and then use causal attention to predict subsequent tokens. In contrast, decoder-only models apply causal attention to all tokens, without distinguishing between input and output. Encoder-only models also do not separate input and output, but they recalculate attention from scratch for each token prediction, performing full attention on all tokens.

prompt design, we included the algorithm code in the prompt so that the LLMs could leverage its knowledge of natural language (see Table 5). We note that the loss was applied only to the answer part of the prompt.

Additionally, we verify the validity of the prompt design used for LLM fine-tuning. To this end, we modify the task from Count3 to Count2 [8] and fine-tune the Llama3-8B using prompts that include algorithmic code, as in the main experiment. As shown in Figure 9, the model successfully learns Count2 with the proposed prompt design, achieving high sequence accuracy. This demonstrates that the model's difficulty in learning Count3 is due to the characteristics of causal decoder-only architecture, rather than an issue with the training strategy, such as the prompt design.

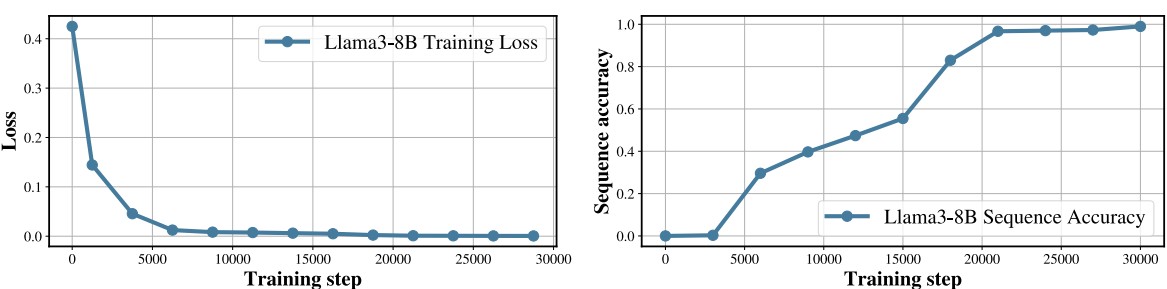

Figure 9: **Results of LLM fine-tuning on** Count2. The Llama3-8B successfully learns Count2 when using the same prompt format as Count3. This demonstrates that the reason LLMs struggle with Count3 is not due to the complexity of the prompt, but rather the characteristics of the decoder-only model.

## C.2 In-context Learning

In this section, we discuss four function classes in the in-context learning experiments (Garg et al., 2022), including the two function classes introduced in the main paper: linear function, sparse linear function, two-layer neural network, and decision tree.

---

[8]$\mathrm{Count2}(x_1, x_2, \ldots, x_n) \coloneqq \left| \left\{ i \in n : x_i + x_n \equiv 0 \pmod{n} \right\} \right| \pmod{n}$.

Table 5: **Example of prompt used for LLM experiments.** We include the code for the algorithm in the prompt to leverage the knowledge of the LLMs.

**Prompt:**
```
def f(x: list[int]) -> int:
    n = len(x)
    count = 0
    for i in range(n):
        for j in range(n):
            if (x[i] + x[j] + x[-1]) % n == 0:
                count += 1
    return count % n
x = [52, 14, 22, 48, 28, 37, 3, 28, 14, 1, 12, 20, 38, 48, 51, 41]
for _ in range(48):
    x.append(f(x))
print(x)
```
What is the output of this code ?

**Output:** [52, 14, 22, 48, 28, 37, 3, 28, 14, 1, 12, 20, 38, 48, 51, 41, 0, 13, 14, 17, 12, 20, 17, 2, 10, 0, 6, 25, 26, 1, 28, 29, 22, 20, 19, 3, 22, 8, 4, 21, 24, 4, 39, 41, 36, 38, 40, 44, 16, 34, 7, 0, 5, 10, 1, 46, 5, 51, 8, 1, 32, 15, 44, 54]

For all function classes, the input $x_i$ is drawn from a Gaussian distribution $N(0, \mathbf{I}_d)$ where $d$ represents the dimension of $x_i$. We provide detailed descriptions of each function class below.

**Linear function.** We consider the class of linear functions $\mathcal{F} = \{f \mid f(x) = w^\top x, w \in \mathbb{R}^d\}$ where $w$ is drawn from a Gaussian distribution $N(0, \mathbf{I}_d)$. Following previous work (Garg et al., 2022), we set $d = 20$ and the number of data points $N$ to 40.

**Sparse linear function.** We also consider a sparse linear function, which is similar to a linear function setup. The difference is that after drawing $w$ from $N(0, \mathbf{I}_d)$, only $k$ randomly selected coordinates are kept, while the remaining ones are set to zero. Following previous work (Garg et al., 2022), we set $k = 3$.

**Two-layer neural network** We examine the class of two-layer ReLU neural networks $\mathcal{F} = \{f \mid f(x) = \mathbf{W}_{(2)}\sigma(\mathbf{W}_{(1)}x), \mathbf{W}_{(2)} \in \mathbb{R}^{1 \times h}, \mathbf{W}_{(1)} \in \mathbb{R}^{h \times d}\}$ where $\sigma(\cdot) = \max(0, \cdot)$ (i.e., ReLU function). We set $h = 100$, $d = 20$, and the number of data point $N$ to 100.

**Decision tree** We consider the class of decision trees represented by full-binary trees of fixed height. In these trees, the leaf node values are drawn from $N(0, 1)$, and the non-leaf nodes are sampled from random integers between 0 and $d$, indicating an index of the input $x$. At each non-leaf node, if the value of the input at the specified index is positive, we move to the right; otherwise, we move to the left. Given an input, we start at the root node and repeat this process until reaching a leaf node. The value of the leaf node becomes the output of the function.

Additional in-context learning experiment results using the four function classes described above are provided in Figure 10. ENTP demonstrates better performance compared to the decoder-only models in linear regression and sparse linear regression, while exhibiting competitive performance in two-layer NN regression and decision tree.

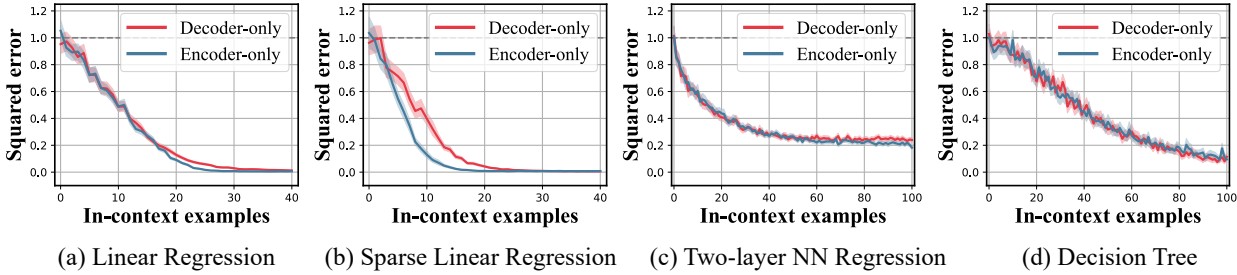

Figure 10: **Additional results of in-context learning experiment.** The encoder-only models demonstrate superior or competitive performance across all function classes compared to the decoder-only models.

## C.3 Addition

We test the sample complexity of encoder-only and decoder-only Transformers using addition tasks, with up to 3-digit numbers. We sample the dataset of all possible 3-digit addition examples using a method similar to the method described in Lee et al. (2023). We start with all 1,000,000 3-digit, 2-digit, and 1-digit addition examples. Then we randomly remove 90% of the 3-digit addition examples, adjusting the ratio of 3-digit to 2-digit examples from around 100:1 to around 10:1. Next we split the data into training, testing, and validation splits, stratified by the number of digits and carries in each addition example. All 1-digit addition examples were put into the training split. Since our models tend to over fit the training dataset, we save the model with the lowest loss on a validation dataset. We test decoder-only and encoder-only Transformers on plain and reversed addition tasks, using between 1.25k to 20k training examples. All sample complexity tests are run with at least 5 different seeds. We test small models, described in Table 6.

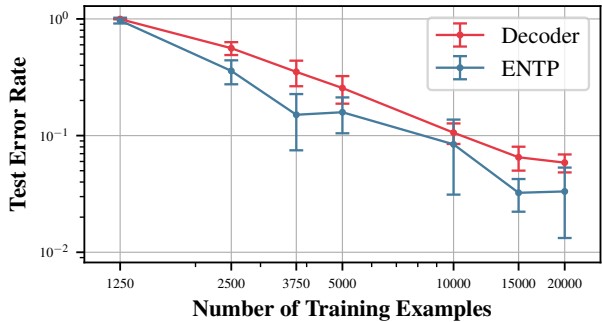

Figure 11: **Addition Sample Complexity.** The train and test datasets include numbers with up to 3 digits. The dataset uses the plain addition format (`$123+456=579$`), unlike the results in Figure 5.

We train Transformers to add larger numbers and evaluate their ability to perform length generalization. Training is performed on numbers with up to 10 digits, while testing extends to numbers with up to 15 digits. Each model is trained on 100,000 examples using the reversed addition format (Lee et al., 2023). The numbers are sampled to ensure equal probability for each length, without duplicates. Consequently, in larger datasets, there are fewer 1-digit addition examples compared to 10-digit ones, as the total number of possible 1-digit examples is smaller. All length generalization tests are run with 3 different seeds. We test Small-Deep models described in Table 6.

## C.4 Model Sizes

In Table 6, we provide the configurations of the Transformer architectures used in the experiments from the main paper.

Table 6: Model specifications.

| Name | Number of Layers | Number of Heads | Embedding Dimension |
|---|---|---|---|
| Small | 3 | 3 | 192 |
| Medium | 6 | 6 | 384 |
| Large | 12 | 12 | 768 |
| Small-Deep | 8 | 2 | 128 |

## C.5 OpenWebText

Table 7 summarizes the results on the OpenWebText dataset across two seeds, and Table 8 lists the hyperparameters used in these experiments.

Table 7: Minimum values of training and validation loss, as well as perplexity, for decoder-only and encoder-only Transformers on the OpenWebText dataset. Results are shown for each seed.

| Model | Train Loss | Validation Loss | Train Perplexity | Validation Perplexity |
|---|---|---|---|---|
| Decoder-only-1 | 4.694 | 4.705 | 109.3 | 110.5 |
| Encoder-only-1 | 4.643 | 4.650 | 103.9 | 104.6 |
| Decoder-only-2 | 4.683 | 4.696 | 108.1 | 109.5 |
| Encoder-only-2 | 4.628 | 4.635 | 102.3 | 103.0 |

Table 8: OpenWebText Hyperparameters

| Parameter | Value |
|---|---|
| warmup_iters | 2000 |
| lr_decay_iters | 600,000 |
| min_lr | 0.00006 |
| max_lr | 0.0006 |
| beta1 | 0.9 |
| beta2 | 0.95 |
| weight_decay | 0.1 |
| block_size | 128 |
| batch_size | 32 |

## D    Implementation of Attention Using $O(D)$ Memory

---

**Algorithm 1** Implementation of Attention Using $O(D)$ Memory

---

**Require:** $q \in \mathbb{R}^{n \times d}$, $k \in \mathbb{R}^{n \times d}$, $v \in \mathbb{R}^{n \times D}$

1: $y_n \leftarrow \mathbf{0}^D$
2: $a \leftarrow \mathbf{0}^D$
3: $b \leftarrow 0$
4: **for** $j = 1, \ldots, n$ **do**
5:     $c \leftarrow \exp(q_n^T k_j)$
6:     $a \leftarrow a + c v_j$
7:     $b \leftarrow b + c$
8: **end for**
9: $y_n \leftarrow \frac{a}{b}$
10: **return** $y_n$

---

## E    $\mathrm{Count3}$ **Algorithms**

---

**Algorithm 2** Algorithm to compute Count3 in $O(n^2)$ time and $O(1)$ space

---

**Require:** length $n$ sequence of integers $(x_1, \ldots, x_n)$

1: count $\leftarrow 0$
2: **for** $i = 1, \ldots, n$ **do**
3:     **for** $j = 1, \ldots, n$ **do**
4:         **if** $(x_i + x_j + x_n) \equiv 0 \pmod{n}$ **then**
5:             count $\leftarrow$ count $+ 1$
6:         **end if**
7:     **end for**
8: **end for**
9: **return** count $\pmod{n}$

---

**Algorithm 3** Algorithm to compute Count3 in $O(n)$ time and $O(n)$ space

---

**Require:** length $n$ sequence of integers $(x_1, \ldots, x_n)$

1: count $\leftarrow 0$
2: table $\leftarrow$ zero-indexed length-$n$ array of 0's
3: **for** $i = 1, \ldots, n$ **do**
4:     $k \leftarrow -x_i \mod n$
5:     table$[k] \leftarrow$ table$[k] + 1$
6: **end for**
7: **for** $i = 1, \ldots, n$ **do**
8:     $k \leftarrow (x_i + x_n) \mod n$
9:     count $\leftarrow$ count $+$ table$[k]$
10: **end for**
11: **return** count $\pmod{n}$

---

# F   RASP Algorithms

In Algorithm 4, 5, and 6, we provide the Python RASP implementation (Zhou et al., 2024) for Count3 and Match3$'$.

---

**Algorithm 4** Count3 RASP Encoder Implementation

---

```python
def g(a, b):
    return a if a < b else a - b

def count_triplets(x):
    idxs = indices(x)

    # set n[i] = len(x) and last_x[i] = x[-1] for all i (only possible with encoder)
    n = sel_width(select(k=x, q=x, pred=true))
    last_x = kqv(k=idxs, q=n - 1, v=x, pred=equals, reduction="mean")

    # g(a, b) is equivalent a % b if 0 <= a < 2 * b
    y = seq_map(n - x, n, g)  # y[i] = -x[i] % n
    z = seq_map(x + last_x, n, g)  # z[i] = (x[i] + x[-1]) % n

    # conut the number of (i, j) such that y[i] == z[j]
    # c = sum(A), where A[i, j] = 1 if y[i] == z[j] else 0
    c = kqv(
        k=full(x, 1),
        q=full(x, 1),
        v=sel_width(select(k=z, q=y, pred=equals)) * n,  # sum(v) = mean(v * n)
        pred=equals,
        reduction="mean",
    )

    # conpute count % n
    c -= idxs * n
    # because count <= n^2, there exists i such that c[i] = count % n or c[i] = n
    # the case c[i] = n is handled by the default value (0) when no keys are selected
    return kqv(k=c, q=n, v=c, pred=lambda a, b: 0 <= a and a < b, reduction="mean")
```

---

---

**Algorithm 5** Match3$'$ RASP Decoder Implementation

---

```python
def has_triplet(x):
    idxs = indices(x)
    first_x = kqv(k=idxs, q=full(x, 0), v=x, pred=equals, reduction="mean", causal=True)

    # use bitmask to compute mod
    y = -x & 127  # y[i] = -x[i] % 128
    z = (first_x + x) & 127  # z[i] = (x[0] + x[i]) % 128

    # max_count[-1] > 0 if there exists (i, j) such that y[i] == z[j]
    max_count = kqv(
        k=full(x, 1),
        q=full(x, 1),
        v=sel_width(select(k=y, q=z, pred=equals)),
        pred=equals,
        reduction="max",
    )

    return tok_map(max_count, lambda a: min(a, 1))  # return 0 or 1
```

---

---

**Algorithm 6** Count3 RASP Decoder COT Implementation

---

```
def count_triplets(x):
    idxs = indices(x)
    n = kqv(k=x, q=full(x, EOS), v=idxs, pred=equals, reduction="min", causal=True)
    n = tok_map(n, lambda a: a if a else -2)
    last_x = kqv(k=idxs, q=n - 1, v=x, pred=equals, reduction="mean")
    seq_len = kqv(k=x, q=x, v=idxs, pred=true, reduction="max", causal=True)

    i = seq_len - n
    j = seq_len - 2 * n
    xi = kqv(k=idxs, q=i, v=x, pred=equals, reduction="max", causal=True)
    xj = kqv(k=idxs, q=j, v=x, pred=equals, reduction="max", causal=True)

    y = (n - xi) % n + 1
    z = (last_x + xj) % n + 1

    y_mask_write = (n <= idxs) & (idxs < 2 * n)
    z_mask_write = (2 * n <= idxs) & (idxs < 3 * n)
    y_mask_read = (n < idxs) & (idxs <= 2 * n)
    z_mask_read = (2 * n < idxs) & (idxs <= 3 * n)

    z_count = sel_width(
        select(
            k=x * y_mask_read,
            q=z,
            pred=lambda a, b: a == b and a != 0,
            causal=True,
        )
    )

    count = kqv(
        k=z_mask_read,
        q=z_mask_read,
        v=n * x * z_mask_read,
        pred=lambda a, b: a & b,
        reduction="mean",
        causal=True,
    )
    ans = count % n

    ans_mask_write = idxs == 3 * n
    eos_mask_write = idxs > 3 * n

    return (
        y * y_mask_write
        + z_count * z_mask_write
        + ans * ans_mask_write
        + EOS * eos_mask_write
    )
```

---

