# OpenReview forum: "Encoder-only Next Token Prediction"
_TMLR — Accepted by TMLR_

### Review · Reviewer_6n5B · 2025-08-13

**Summary Of Contributions:**

The authors studied an encoder based next token prediction transformer, where Figure 1 provides a clear picture of the architecture and method. The authors demonstrated that encoder and decoder architectures have distinct representation powers theoretically, but more importantly identified a task - count3 - that only encoder based transformers seem to be able to perform. There is also additional theoretical results on the complexity of such tasks as well, along with further experimental results.

**Audience:**

Yes

**Audience Explanation:**

While the encoder based architecture is less efficient computationally, it remains interesting to study the behaviour as a theoretical tool to understand the complexity of certain language type tasks like count3, as well as the limitations of decoder only architectures. For one, it is surprising that not only decoder based architectures fail at count3 but also encoder succeeds. This itself is a sufficiently interesting read for me, and I believe it should be interesting to many other audience members of TMLR.

**Claims And Evidence:**

Yes

**Claims Explanation:**

All theoretical results came with associated proofs, and all of them after a high level read appear sound, albeit I did not check super carefully about potential mistakes.

Conjecture 6.1, while not proven, seems highly plausible given the authors provided Algorithm 2 and 3, which does indeed appear optimal. It also appears potentially challenging and somewhat outside the scope of this paper to prove the conjecture, so it is appropriate the authors left it as unproven.

Figure 3 and 4 provides very convincing empirical evidence that count3 is a solvable (albeit not necessarily trivial) task for the encoder based architecture, while simultaneously showing that the decoder based architecture cannot solve the task easily (note the saturation in Figure 3 left at step 10k-12k for decoder architectures).

**Requested Changes:**

Most of the paper is well written, except I would like a clearer mathematical description of the difference between decoder-only and encoder-only next token prediction. Figure 1 is quite clear, but the structure is not very obvious in the mathematical setup. Perhaps the authors can consider writing down literally the equations corresponding to Figure 1 and highlighting the difference.

---

> ### Author Response · Authors · 2025-09-18
>
> We thank the reviewer for the encouraging feedback, highlighting (i) the significance of the count3 task, (ii) that the paper is well written and (iii) the broader interest to the TMLR audience. Following the reviewer's suggestion, we have added a new appendix section (A.4) that explicitly derives the computation of a two-layer encoder and decoder respectively, highlighting the differences.

---

### Review · Reviewer_W7Re · 2025-08-28

**Summary Of Contributions:**

### Summary

The paper analyzes how causal, decoder-only Transformers differ from bidirectional, encoder-only Transformers in sequence generation tasks. There are both theoretical and empirical arguments made here.

1. First is **a theoretical result showing that in terms of expressivity, counter-intuitively, neither architecture subsumes the other.** (Note that this result is existential: it constructs a single instance of a function that one can represent, but the other cannot match exactly.)
   - This is a surprising result since one would typically think that the bidirectional architecture is strictly superior in expressivity.
   - Note that if one were to consider whether one can at least approximate the other inexactly, then it becomes clear that the bidirectional model can approximate any decoder model, after all.
   - The result admittedly holds under some specific limitations/assumptions.

2. The next main result is **the theoretical analysis of an autoregressive sequence-generation task** called Count3 (which is repurposed from an existing task in literature).
   - Here, it is shown that a causal transformer would require layers growing with the sequence length whereas a bidirectional one can do with constant number of layers.
   - Note that the theorem holds under an assumed conjecture (a reasonable one).

3. Finally, **two sets of empirical results** demonstrating the superior performance of the encoder-only architecture:
    - First, are on the Count3 task demonstrating a gap in performance on some decoder-only, prefix-based decoder-only, and some medium-sized and large BERT-based encoder models.
    - Other empirical results on addition (length-generalization, sample complexity), in-context learning, natural language tasks (commonsense reasoning, classification)


## Strengths

1. The paper analyzes a good scientific question. Even though
    - bidirectional attention is expensive and currently impractical, and
    - it may be unsurprising that it must have some superior expressivity or capabilities or sample complexity,

this gap must be quantified and studied scientifically. We need to know what the headroom is, and have insight into when/what type of task this becomes relevant. The paper addresses this.

2.  The analysis is clean and understandable. The examples studied are easy to follow and illustrative. There is no unnecessary mathematical obfuscation. I found this a pleasure to read.

3. The paper encompasses both theoretical and empirical analyses, and these analyses complement each other well. I contrast this against a paper that is focused on proving theorems, or a paper that is a collection of empirical results.

4. Importantly, the paper is honest about its limitations where applicable.


## Weaknesses

There is an obligatory list of weaknesses I could state:
- I would say that the overall claim is qualitatively expected.
- Furthermore, the theoretical constructs (e.g., the Count3 task) may appear a bit artificial.

But I would state these as acceptable weaknesses for the chosen direction, and for the style of accessible result the paper aims for.

However, I require some major clarifications before acceptance.


## Major Clarification question 1

**Do the experiments in Sec 6 report a gap in training accuracy or test accuracy? (e.g., Fig 3 right?)**  (There’s a similar clarity issue in Fig 7) **This is an important question because it somewhat contradicts my expectation.** Let me explain.


The theorem in Sec 6 presents an expressivity gap in a task (not in the ability to learn with gradient descent). It’s possible that when it comes to _learning_, both architectures fail (thus, the gap goes away). Now, it’s clear that the experiments demonstrate a gap in training loss (this is still an expressivity gap). But is it also demonstrating a test accuracy gap?

This bothers me because I would find it surprising if this task can be learned in the first place with any architecture, without any step-wise supervision or some form of curriculum (i.e., train on a mixture of Count3 with n=1, 2, 3…). The task involves composing multiple subroutines, which is generally a hard learning problem. See for example Fig 2 in [1] who use a mixture of various task complexities.


## Major Clarification question 2

**The claim at the end of Sec 4 doesn't seem justified:**  At the end of Sec 4, the paper promises that the converse approximation result does not hold. I.e., “there exist some ENTP models that decoder-only
Transformers cannot approximate.” But the theorems still talk about exactly fitting the counts. Could you clarify? Perhaps this claim needs editing.


## Minor clarification question 3

**A jump between the empirical and theoretical claims:** The theoretical results are about expressivity; but some of the later empirical results involve sample complexity (which relates to learning). There seems to be a jump here that the paper is a bit casual about, and does not make the reader aware of. I would like to hear more from the authors about this: is there intuition for the sample complexity gap?

[1]: "Transformers, parallel computation, and logarithmic depth", Clayton Sanford, Daniel Hsu, and Matus Telgarsky https://arxiv.org/pdf/2402.09268

**Additional Comments:**

Some suggestions:

1. Given that Theorem 4.2 is a bit counterintuitive (it says that ENTP does not strictly subsume causal decoder-only models), the reader would benefit from proof intuition here.
2. The introduction of the task in Eq 6 should immediately let the reader know that the task requires the model generating multiple tokens or equivalently that the equality should hold for various values of m ranging upto some length n.  My initial impression was that we only need one equality to hold at position n and so the proof confused me a bit.
3. Under 6.1, the text reads “sequences generated from Equation (6)” I’d add “under a distribution defined as follows”, because Eq 6 does not fully describe a generative process.
4. In Conjecture 6.i, it wasn’t immediately clear what “with access to x_n” meant until I read the proof.
5. In Section 7, I’d reassure the reader that the two models you compare are of similar sizes. (I had to chase the links down to verify what models were used.)

**Audience:**

Yes

**Audience Explanation:**

The paper presents an accessible set of quantitative and analytical results demonstrating the effect of a fundamental aspect of language models (the directionality of the attention mechanism). The results are both theoretical and empirical in nature, and would thus be of  significant interest to a broad range of readers in the TMLR community.

**Broader Impact Concerns:**

Not applicable.

**Claims And Evidence:**

Yes

**Claims Explanation:**

I checked the main paper's proofs which look correct. There are also empirical results which corroborate the claims.

**Requested Changes:**

Please provide clarifications to my questions, and update the paper accordingly. Importantly, it is unclear what exactly the experiments on Count3 demonstrates -- this would help me assess the relevance of these experiments to the main claim.

---

> ### Author Response · Authors · 2025-09-18
> **Author Response**
>
> We thank the reviewer for the thoughtful feedback, and for acknowledging that (i) our work analyzes an important question of "headroom" between encoder and decoder models, (ii) the analysis is clean and understandable, free from unnecessary mathematical obfuscation, and (iii) it combines theoretical and empirical analyses that complements each other well.
>
> > **Do the experiments in Sec 6 report a gap in training accuracy or test accuracy?**
>
> In our Count3 experiments, we generate sequences from random seeds. With our specific setup, seeds contain 16 integers between 0 and 63, so there are $64^{16} \approx 1.16 \times 10^{77}$ possible seeds. Thus, the chance of significant duplication among the $1.28 \times 10^7$ seeds used for training and evaluation is negligible. Consequently, the reported results effectively correspond to "test" loss and accuracy. We clarify this in Section 6.2 of the revised paper.
>
> As demonstrated by Zhou et al. [1], Transformers are capable of learning tasks that can be performed by simple RASP programs. In Appendix F, we provide a RASP program for Count3 with ENTP. This suggests it is reasonable to assume that ENTP can learn Count3, provided sufficient training data.
>
> When training on Count3 with $n=64$ and a seed of 16 integers, the model is exposed to sequences with $n \in [17, 64]$, since every non-seed integer is generated using the Count3 function. In this way, the training process inherently provides a curriculum over multiple values of $n$.
>
> [1] Zhou et al. (2023). What Algorithms Can Transformers Learn? A Study in Length Generalization. https://arxiv.org/abs/2310.16028
>
> > **The claim at the end of Sec 4 doesn't seem justified: At the end of Sec 4, the paper promises that the converse approximation result does not hold.**
>
> > Claim: There exist some ENTP models that decoder-only Transformers cannot approximate.
>
> Lemma 6.2 concerns exactly fitting a discrete function such as Count3. Many different continuous Transformer realizations can interpolate such a function. This is in contrast to Section 4 which is explicitly about exact replication and approximation of continuous functions mapping sequences of vectors in $\mathbb R^d$.
>
> If no decoder-based language model can exactly replicate a given ENTP-based language model (the argmaxes of their outputs agree), then there must also be a limit on how closely a decoder can approximate its output embeddings, assuming there is a nontrivial gap between the largest and second largest logit values (which holds for our ENTP models for the Count3 task).
>
> Thus, since our lemmas show that no decoder-only Transformer can exactly solve the Count3 task, while an ENTP model can, the observation above implies that a decoder cannot approximate arbitrarily well an ENTP model that successfully solves Count3. We clarify this in Section 6.2 of the revised paper.
>
> > **The theoretical results are about expressivity; but some of the later empirical results involve sample complexity (which relates to learning).**
>
> We agree that our theoretical results are about expressivity, while some empirical results involve sample complexity. The connection is that if the simplest algorithm for a task can be expressed by ENTP but not by a decoder (for example, if the algorithm relies on non-causal internal states) then the task may be easier for ENTP to learn. In such cases, ENTP’s better alignment with the task can lead to lower sample complexity and stronger generalization.

---

### Review · Reviewer_LcnL · 2025-09-06

**Summary Of Contributions:**

This paper challenges the conventional reliance on decoder-only Transformers for next-token prediction and proposes Encoder-only Next Token Prediction (ENTP) as an alternative. The authors offer a comprehensive comparison between encoder-only and decoder-only architectures in terms of expressive power and computational complexity, both theoretically and empirically. A new function, Count3, is introduced to demonstrate settings where ENTP models succeed but decoder-only Transformers provably cannot, and empirical results across several representative tasks—including arithmetic, in-context learning, and language modeling—are presented to support the main claims.

**Audience:**

Yes

**Audience Explanation:**

As far as I am concerned, researchers in the TMLR community may feel it is intreseting to explore new-architecture of LLMs beyond GPT-style transformer decoders. Though transformer encoder is not a brand-new architecture (e.g., PLMs in BERT-family), I think it is interesting to see the potential of transformer encoder in the next-token prediction language modeling task.

**Claims And Evidence:**

Yes

**Claims Explanation:**

Specifically,
- The paper offers a careful and detailed theoretical analysis of the expressive power and complexity differences between encoder-only and decoder-only Transformers, supported by explicit theorems (Section 4, Page 4).
- A central technical novelty is the introduction and empirical investigation of the Count3 task, which is motivated and analyzed in-depth both algorithmically and experimentally (Section 6, Pages 6–9). The discussion is connected to existing complexity arguments, strengthening the theoretical case.
- Empirical results systematically compare ENTP with alternative architectures across a wide suite of tasks. On the Count3 setup, Figure 3 (Page 9) provides compelling evidence that ENTP learns the task efficiently while decoder-only models (including large LLMs as in Figure 4 on Page 9) fail entirely. Table 2 (Page 10) supports that Match3’, a similar but easier function, is learnable by both, helping calibrate the findings.
- On small-scale language modeling, arithmetic, and in-context learning benchmarks, Figures 5–7 (Pages 10–11) and Tables 3–4 (Page 12) show that ENTP achieves competitive or superior results on sample complexity, length generalization, and real-word tasks compared to decoder-only Transformers.

**Requested Changes:**

- While the existential theorems (Theorem 4.1/4.2, Page 4) and complexity analyses provide insight, the generality relies on idealized settings (unbounded token/embedding dimensions, arbitrary input domains), reducing immediate practical impact. The circumstances where ENTP's theoretical benefits transfer to real world—especially with finite token sets and typical sequence lengths—are only partially discussed (Page 4 and Page 7, top). For instance, the inability of decoders to approximate certain ENTP-expressible functions is shown in a very specific and contrived setting, and does not rule out that for most realistic language modeling objectives, the gap is negligible.
- The ENTP architecture as presented carries severe computational overhead (Section 5, Table 1, Page 6), scaling as $O(n^3)$ per sequence for time (vs $O(n^2)$ for decoders) and increasing space complexity. While this is acknowledged and framed as a tradeoff, the lack of systematic exploration of this bottleneck in practice limits ENTP's applicability, and no mitigation strategies are attempted. Readers interested in scaling would likely be disappointed by the absence of even preliminary scaling/efficiency experiments or ablations.
- Most empirical results focus on small or synthetic settings. For actual natural language modeling, Table 3 (Page 12) shows only a slim improvement in validation loss/perplexity, and these are over a relatively small OpenWebText subset with mid-size models. Notably, there are no controlled ablations or extensive benchmarking against strong masked language modeling (MLM) variants or encoder-decoder architectures, nor against recently proposed improvements for next-token prediction efficiency (such as those discussing next-sentence or segment prediction objectives).
- The impact of ENTP's design on downstream transfer, fine-tuning, compositionality, or large-scale context understanding (beyond length generalization on addition) is only superficially explored. Figures 5–7 and Table 4 hint at outperformance but lack detail on statistical significance, robustness (e.g., random seeds, repeatability), and comparative performance with state-of-the-art models.
- Would hybrid architectures (e.g., partial causal masks or adaptive masking) bridge the gap between ENTP's expressiveness and decoder efficiency?

---

> ### Author Response · Authors · 2025-09-18
> **Author Response (1/2)**
>
> We thank the reviewer for the thorough feedback, and for acknowledging that (i) our work provides a thorough theoretical analysis and (ii) we provide a systematic empirical comparison of ENTP against alternative architecture.
>
> We also wish to highlight positive feedback from the other reviewers: (i) our work addresses a meaningful scientific question with rigorous quantification, and (ii) although ENTP is less computationally efficient, it serves as a valuable tool for clarifying the limitations of decoder-only models and studying the complexity of language-types.
>
> > **On the scope and motivation of our work**
>
> We would like to emphasize that the **primary objective of our study is to explore scientific questions about Transformers, rather than to advocate for ENTP as a practical language modeling solution given current hardware limitations**. Our results aim to characterize the expressive power and learning behavior of encoder-only next-token prediction (ENTP) relative to decoder-only models. We see this as an inquiry into the capabilities and limitations of Transformer variants. While we do acknowledge computational limitations of ENTP, our intention is not to propose ENTP as an immediately scalable alternative, but rather to use it as a scientific probe into how architectural choices affect expressivity, learnability, and generalization.
>
> > **On the generality of the theoretical results**
>
> We agree with the reviewer that Theorems 4.1 and 4.2 are proven in idealized settings. Our goal in presenting these theorems is not to claim direct practical applicability, but rather to establish separation results: namely, that encoder-only and decoder-only Transformers are not strictly more expressive than one another, and in fact capture different sets of functions. These theorems are valuable because they show that simply changing the attention mask can fundamentally alter the class of causal functions a Transformer architecture can or cannot compute.
>
> > **On the relevance of synthetic tasks like Count3**
>
> Synthetic tasks are an established and valuable tool for studying the algorithmic capabilities and limitations of Transformers. They provide controlled environments that isolate specific reasoning skills, making it possible to uncover structural properties of architectures that would be difficult to identify in large-scale natural language experiments. Recent work has shown the importance of such tasks for understanding and improving Transformer design, including Physics of Language Models (Allen-Zhu et al., 2024), Teaching Arithmetic to Small Transformers (Lee et al., 2023), Transformers Can Do Arithmetic with the Right Embeddings (McLeish et al., 2024), What Algorithms Can Transformers Learn? (Zhou et al., 2023), Length Generalization in Arithmetic Transformers (Jelassi et al., 2024), Looped Transformers for Length Generalization (Fan et al., 2024), What Can Transformers Learn In-Context? (Garg et al., 2023), and Unveiling Transformers with LEGO (Zhang et al., 2022).
>
> We designed Count3 as a minimal synthetic task that requires exploitation of all pairs of inputs for each output, making it a simple yet rigorous probe of architectural limits. Importantly, Count3 can be studied with short sequences (64 tokens) which keeps ENTP experiments computationally feasible while still exposing fundamental differences in expressivity. The significance of our findings lies in the fact that the performance gap is not merely a gap present when matching parameter counts or compute budgets. Decoder-only models fail to learn Count3 even at large scales. In contrast, small ENTP models succeed, demonstrating a structural limitation of decoders.

---

> > ### Author Response · Authors · 2025-09-18
> > **Author Response (2/2)**
> >
> > > **On the small gap in real-world language modeling performance**
> >
> > Our experiments on real-world language modeling were limited by computational constraints and thus are not the focus of this work. The amount of training data we could use restricted our pretrained models to GPT-2–level tasks, and unlike synthetic tasks such as Count3 and addition, we were unable to train these models to completion. These limitations likely prevented our ENTP models from fully leveraging the architecture’s expressivity.
> >
> > > **On the robustness of experimental results**
> >
> > For addition (Figures 5 and 6), results are averaged over three seeds, as described in Appendix C.3. In-context learning (Figure 7) and language modeling (Table 3) originally relied on a single seed due to the high cost of training ENTP in those settings. Following the reviewer’s helpful feedback, we conducted an additional OpenWebText pretraining run with a different random seed and incorporated the updated results. Based on this additional pretrained model, we also carried out further validation on TinyWinoGrande (Table 4) under the same zero-shot greedy decoding setup without fine-tuning. CLUTRR experiments (Table 4) were fine-tuned and evaluated with three seeds, as described in Section 7.3. Although resource constraints currently limit the number of seeds, in the final version we will provide additional multi-seed results, including for in-context learning, to more clearly demonstrate the robustness of our findings.
> >
> > In addition, we validated our training setup for Count3 experiments by testing alternative functions while keeping the rest of the training setup unchanged. Specifically, we verified that decoder-only models can reliably learn simpler tasks such as Count2 and Match3’, achieving high accuracy. This demonstrates that the decoder’s failure on Count3 arises from the inherent difficulty of the function itself rather than shortcomings in our training procedure.
> >
> > > **Note on hybrid architectures**
> >
> > In our Count3 experiments (Figure 3) we test Prefix decoder-only models (Raffel et al., 2020; Wu et al., 2021), which perform noncausal attention for the prefix portion of the sequence. We find that they slightly outperform decoder-only models, but still fail to learn Count3. We leave further study of hybrid architectures and strategies for mitigating complexity to future work.

---

### Decision · Action_Editor_TuRz · 2025-10-05

**Recommendation:** Accept as is

**Audience:**

Yes

**Audience Explanation:**

All reviewers agreed this could be of interest to members of the TMLR community, given that it presents a novel perspective on a core task underlying current state-of-the-art models (next-token prediction).

**Claims And Evidence:**

Yes

**Claims Explanation:**

All reviewers agreed the claims were supported by convincing evidence, both empirical and theoretical.